# SPREAD-GS: Scale-Progressive Representation Extraction and Detailing for 3D Gaussian Splatting

## Abstract

3D Gaussian Splatting (3DGS) has recently shown remarkable capability for high-fidelity scene reconstruction. However, its potential for object recognition remains under-explored. Existing approaches often extract 2D features from multi-view images and embed them into 3DGS, which limits the joint use of 3DGS geometric and appearance information. To fully exploit both the structural and fine-grained details encoded in 3DGS, we propose Scale-Progressive Representation Extraction And Detailing for 3D Gaussian Splatting (SPREAD-GS), a framework for object classification that combines scale-aware sampling with detail-preserving feature propagation. SPREAD-GS has two key modules: Scale-Progressive Sampling (SPS), generating multi-scale subsets by progressively narrowing the visible region, and SpreadNet, encoding these subsets and propagating details across scales through noise-augmented feature upsampling. On the texture-rich MACGS dataset, SPREAD-GS achieves 93.93% overall accuracy, improving the SOTA by 2.02%. On the geometry-centric ModelNet40GS, it matches the SOTA while significantly reducing parameters. These results demonstrate the effectiveness of scale-progressive sampling and detail-preserving feature propagation for 3DGS recognition. Our code is available at https://anonymous.4open.science/r/noname-64BE.

## 1 Introduction

Understanding 3D objects is a fundamental problem in computer vision, with a wide range of applications in scenarios such as autonomous driving, robotics, and virtual reality. Over the past decade, researchers have explored various 3D representations, including voxels, meshes, point clouds, and neural radiance fields (NeRF)(Mildenhall et al., 2021). Each of these representations has its own advantages: voxels provide regular geometric information, meshes can precisely depict surface geometry, point clouds are lightweight and flexible, and NeRF supports high-fidelity rendering. In recent years, 3D Gaussian Splatting (3DGS) (Kerbl et al., 2023) models scenes with Gaussians primitives and spherical harmonic coefficients, enabling real-time rendering while preserving appearance details. This representation opens new opportunities for 3D object understanding.

Before 3DGS, extensive research on 3D object recognition focused on voxel(Maturana & Scherer, 2015), mesh (Hanocka et al., 2019), and point cloud (Qi et al., 2017a) representations. In contrast, recognition with 3DGS remains largely unexplored. A common strategy (Qin et al., 2024) (Guo et al., 2024) is to extract image-level features via large-scale vision models and embed them into 3DGS, but this often loses 3D structural information because images inherently lack 3D geometry. Subsequent work (Ma et al., 2025) (Zhang et al., 2025) has introduced novel perspectives on 3DGS recognition, viewing 3DGS as an extension of point clouds and processing it with point cloud networks. However, it concatenates Gaussian attributes—including position, scale, rotation, opacity and colour —into a single feature vector and feeds it directly into point cloud networks. While 3DGS can be viewed as a discrete representation analogous to point clouds, allowing point cloud networks to get global structure, its richer per-primitive attributes encode finer local details. Directly concatenating these attributes into a single vector for point cloud networks risks losing this localized information. To address this deficienc, our method introduces a novel sampling strategy and a hierarchical architecture designed to encode global structures and local details within 3DGS.

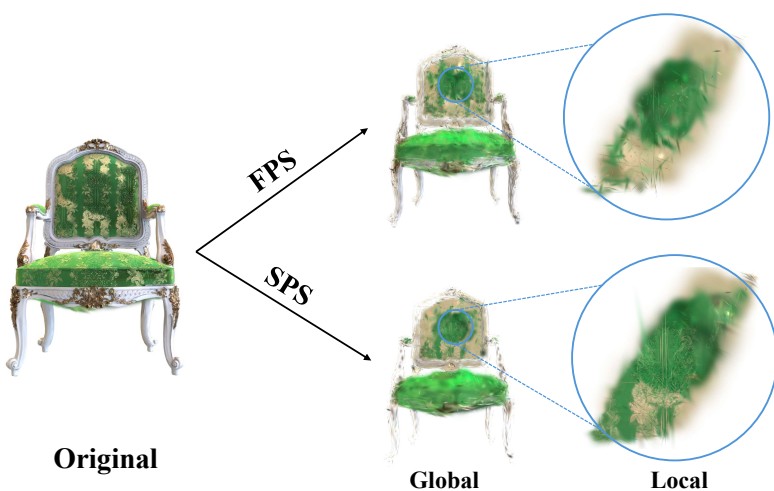

Fig. 1: Comparison between FPS and our SPS. Under the same number of sampled points, FPS produces a uniform distribution across the entire shape, which may dilute central details. In contrast, SPS progressively samples within FoV cones, preserving the global structure while retaining finer details near the viewing center.

In this work, we propose Scale-Progressive Representation Extraction And Detailing for 3D Gaussian Splatting (SPREAD-GS), a framework for 3DGS classification. We design a Scale-Progressive Sampling (SPS) strategy that generates multi-scale subsets by progressively narrowing the visible region, capturing both global structure and progressively finer details to better utilize 3DGS representations. Fig. 1 illustrates the difference between SPS and standard Farthest Point Sampling (FPS) (Qi et al., 2017b). Based on this, we build a hierarchical network, SpreadNet, unlike traditional networks such as PointNet++ (Qi et al., 2017b) and DGCNN (Wang et al., 2019), which encodes layered inputs and propagates details across scales through noise-augmented feature upsampling, effectively integrating structural and appearance cues. On the detail-rich MACGS dataset, SPREAD-GS achieves 93.93% overall accuracy (OA), surpassing the previous state-of-the-art (SOTA) by 2.02%, and on the structure-focused ModelNet40GS (Ma et al., 2025) it reaches 91.87% OA, exceeding the previous SOTA by 0.32%, reducing memory usage by nearly 10 times. These results indicate that SPREAD-GS is most effective when combining geometric and appearance information, which explains its strong performance on texture-rich datasets and modest improvement on purely geometric ones.

To sum up, our contributions include: (1) we introduce SPREAD-GS, a new classification network directly modeling native 3DGS attributes; (2) we design SPS and SpreadNet for joint multi-scale modeling of structure and appearance; (3) we achieve significant improvements over point cloud baselines on MACGS and ModelNet40, validating the effectiveness of our approach.

## 2 RELATED WORK

### 2.1 3D REPRESENTATION LEARNING

Early explorations of 3D semantic understanding investigated multi-view (Su et al., 2015; Wei et al., 2020), voxel-based (Maturana & Scherer, 2015; Wu et al., 2015), and mesh-based (Hanocka et al., 2019; Gong et al., 2019) representations. These approaches leveraged 2D images or regular structures but often suffered from high memory cost, information loss, or strict mesh requirements, motivating the shift toward more flexible point-based paradigms.

Point clouds provide a direct representation of 3D geometry without voxelization or rendering, enabling more efficient learning. PointNet (Qi et al., 2017a) pioneered end-to-end modeling of raw point sets, and PointNet++ (Qi et al., 2017b) extended this idea with hierarchical local feature learning. DGCNN (Wang et al., 2019) introduced dynamic graph convolutions to capture local geometric relations, while subsequent methods improved robustness and efficiency (Yang et al., 2024; Ma et al., 2022; Liang et al., 2024). Transformer-based architectures (Zhao et al., 2021; Wu et al., 2022;

2024) further incorporated global dependencies. Together, these advances establish point clouds as a powerful and flexible paradigm for hierarchical 3D reasoning.

## 2.2 3D GAUSSIAN SPLATTING FOR SEMANTIC UNDERSTANDING

With the recent success of 3D Gaussian Splatting (3DGS) in high-fidelity scene representation (Kerbl et al., 2023; Liu et al., 2024), several works have explored its semantic understanding. One line leverages 2D-driven priors from large vision models. LangSplat (Qin et al., 2024) projects 2D CLIP (Radford et al., 2021) features into 3D Gaussian space for open-vocabulary localization. SAGA (Cen et al., 2025) embeds SAM-derived 2D features into Gaussians via contrastive learning. FMGS (Zuo et al., 2025) distills multi-view 2D foundation model features into 3D Gaussians for fast 3D semantic reconstruction. CLIP-GS (Liao et al., 2024) projects multi-view 2D CLIP features into 3D Gaussians for open-vocabulary 3D segmentation. FlashSplat (Shen et al., 2024) lifts multi-view 2D masks to 3D Gaussians via a linear program for fast, robust segmentation. Semantic Gaussians (Guo et al., 2024) leverages 2D superpixels for semantic annotation. Feature3DGS (Zhou et al., 2024) enables distillation of high-dimensional 2D foundation model features into each Gaussian. Gaussian Grouping (Ye et al., 2024) leverages 2D image features from large vision models to construct identity encodings for 3D Gaussians. SplatTalk (Thai et al., 2025) encodes multi-view 2D images into vision–language features, embeds them into 3D Gaussians, and extracts 3D tokens for zero-shot visual question answering. While effective, these methods remain tied to 2D features, introducing ambiguities and scale inconsistencies when projected into 3D.

Subsequent work emphasizes native 3D reasoning in Gaussian Splatting. ShapeSplat (Ma et al., 2025) introduces a large-scale dataset of 3DGS and a masked autoencoder (Gaussian-MAE) for self-supervised pretraining to improve 3D classification and segmentation. MACGS (Zhang et al., 2025) concatenates Gaussian attributes—position, scale, rotation, opacity—into point cloud representations, which are then fed to point-cloud networks trained for object classification. However, they largely treat 3DGS as point clouds and ignore its ability to encode fine-grained local details. To address this limitation, we propose SPREAD-GS, which exploits the rich intrinsic attributes of 3DGS, and capturing multi-scale structural and detailed information to more effectively leverage 3DGS representations for object classification.

## 3 PRELIMINARY

3D Gaussian Splatting (3DGS) (Kerbl et al., 2023) represents a 3D scene or object as a collection of anisotropic Gaussian primitives, enabling differentiable and efficient rendering. Each primitive is parameterized by a center $x_i \in \mathbb{R}^3$, a scaling factor $s_i \in \mathbb{R}^3$, a rotation quaternion $q_i \in \mathbb{R}^4$, an opacity $\alpha_i \in \mathbb{R}$, and spherical harmonic coefficients $sh_i \in \mathbb{R}^d$. The full representation is

$$\mathcal{G} = \{g_i\}_{i=1}^N, \quad g_i = \{x_i, s_i, q_i, \alpha_i, sh_i\}. \tag{1}$$

The covariance of the $i$-th Gaussian is defined via its scale and rotation:

$$\Sigma_i = Q_i S_i S_i^T Q_i^T, S_i = \text{diag}(s), Q_i = R(q_i) \tag{2}$$

where $R(q_i) \in SO(3)$ is the rotation matrix derived from $q_i$. Each Gaussian thus corresponds to an ellipsoidal density:

$$\mathcal{G}_i(x) = \alpha_i \exp\left(-\tfrac{1}{2}(x - x_i)^T \Sigma_i^{-1}(x - x_i)\right). \tag{3}$$

This parameterization is expressive, differentiable, and avoids issues of directly optimizing unconstrained covariance matrices, as $\Sigma_i$ is guaranteed positive semi-definite by construction. In practice, the set $\mathcal{G}$ compactly encodes both geometry and appearance, and serves as the input representation for our scale-progressive sampling and hierarchical network design.

## 4 METHOD

In this section, we introduce SPREAD-GS, a framework for classifying fine-grained objects represented by 3D Gaussian Splatting. The process is shown in Fig. 2, the pipeline consists of three stages: (1)input Gaussians are processed by the Scale-Progressive Sampling (SPS) strategy, which

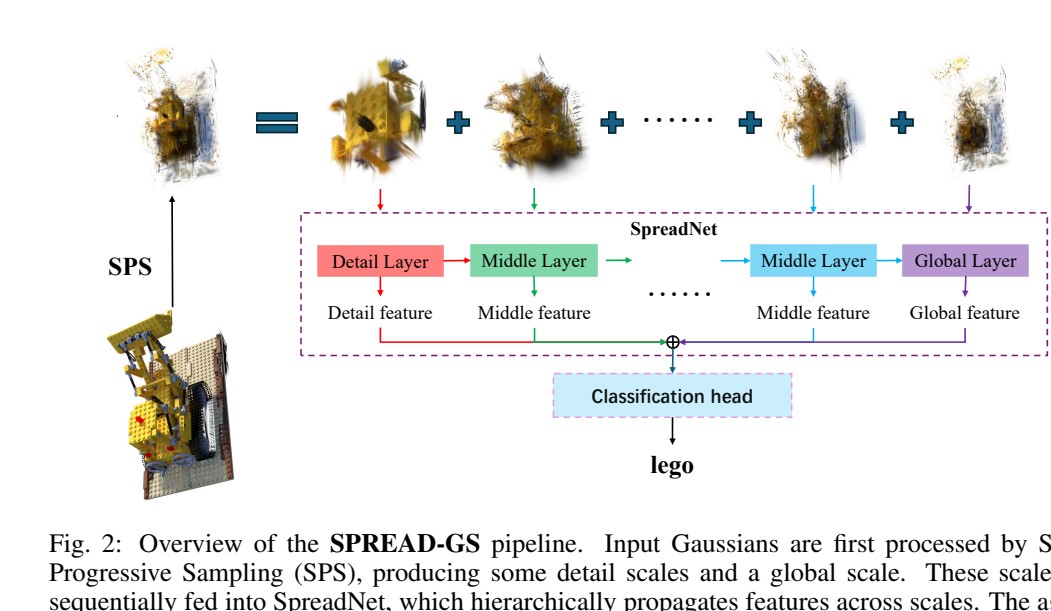

Fig. 2: Overview of the **SPREAD-GS** pipeline. Input Gaussians are first processed by Scale-Progressive Sampling (SPS), producing some detail scales and a global scale. These scales are sequentially fed into SpreadNet, which hierarchically propagates features across scales. The aggregated features are then passed to a standard classification head to produce the final object class.

generates multiple detail scales together with a global scale; (2) these scales are fed into Spread-Net, a hierarchical extractor aligned with the SPS design, where additional middle layers can be added if more scales are used; (3) the aggregated features are passed to a conventional classification head. The classification head follows standard practice, while our main contributions lie in SPS and SpreadNet, detailed in Sec. 4.1 and Sec. 4.2.

## 4.1 SCALE-PROGRESSIVE SAMPLING

Extracting features from fine-grained 3DGS objects is challenging because they contain a large and variable number of Gaussian primitives, so processing them all is currently computationally expensive in practice. A common solution is Farthest Point Sampling (FPS)(Qi et al., 2017b), which selects points farthest from the already chosen ones to ensure uniform spatial coverage, but it will discard fine-grained details. To address this, we propose Scale-Progressive Sampling (SPS). The core idea of SPS is to progressively restrict the spatial sampling region so that the number of Gaussians within each region becomes comparable to the number of points to be sampled; when these counts are comparable, a sampler can preserve local details. During the progressively narrowing process, earlier (larger) regions yield samples that emphasize global structure, while later (smaller) regions permit denser sampling that preserves fine local information; combining these regimes balances global and local cues.

Concretely, SPS sequentially reduces the camera field of view(FoV), collects Gaussians within the visible area, and applies a standard sampler (e.g., FPS) within each region. Formally, given the full set of Gaussians: $\mathcal{G} = \{g_i = (x_i, s_i, q_i, \alpha_i, sh_i) \mid i = 1, \ldots, N\}$ as defined in Eq. (1), the SPS paradigm produces a sequence of subsets $\mathcal{G}^{(0)}, \mathcal{G}^{(1)}, \ldots, \mathcal{G}^{(\ell)}$, corresponding to progressively narrower camera FoV constraints. The overall process is shown in Fig. 3.

SPS proceeds in three steps:

1. **Global sampling**: Apply sampling to the entire set $\mathcal{G}$ to obtain an initial coarse subset $\mathcal{G}^{(0)}$.

2. **Progressive frustum sampling**: At each iteration, a FoV angle $\theta^{(\ell)}$ is chosen to define the visible subset of Gaussians:

$$\mathcal{G}_{\theta^{(\ell)}} = \{g_i \in \mathcal{G} \mid x_i \in \mathcal{F}(\theta^{(\ell)})\}, \tag{4}$$

which restricts the candidates to those lying inside the current frustum and $\mathcal{F}(\theta^{(\ell)})$ denotes the camera frustum with FoV angle $\theta^{(\ell)}$. Sampling is then applied within $\mathcal{G}_{\theta^{(\ell)}}$ to obtain a representative subset $\mathcal{G}^{(\ell)}$, ensuring spatial uniformity and preserving both global and local

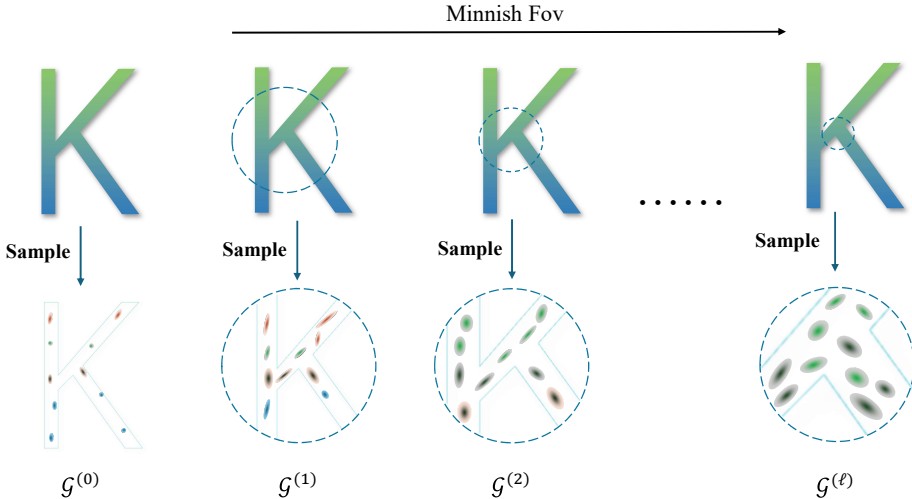

Fig. 3: Illustration of Scale-Progressive Sampling (SPS). (1) SPS first samples the complete object to obtain $\mathcal{G}^{(0)}$ (global structure), then iteratively narrows the view to produce $\mathcal{G}^{(1)},\ldots,\mathcal{G}^{(\ell)}$. (2) As the FoV narrows, the number of visible Gaussians approaches the sampling count, and the final $\mathcal{G}^{(\ell)}$ retains fine-grained local details without loss.

cues. Finally, the FoV is progressively narrowed, $\theta^{(\ell+1)} < \theta^{(\ell)}$, and the process is repeated on the reduced frustum.

3. **Stopping criterion**: The process terminates when the number of Gaussians within the frustum becomes sufficiently small, such that sampling preserves meaningful local information. In implementation, we stop sampling once the number of Gaussians before and after sampling remain within the same order of magnitude ($\leq 10\times$ difference), as validated in the Appendix A.0.1.

In practice, SPS can introduce a "central bias", where points near the frustum center are preferentially sampled. To alleviate this, we apply a random shift to the normalized point cloud before SPS, ensuring that local Gaussians across the entire frustum can be effectively sampled. The quantitative impact of this augmentation is reported in the Appendix A.0.2.

## 4.2 Scale-Progressive Representation Extraction And Detailing Network

Scale-Progressive Representation Extraction And Detailing Network (SpreadNet) is a hierarchical network that processes the multi-scale subsets generated by SPS, progressively fusing information across scales to capture both fine local details and global structure in 3DGS representations. We assume that local structural and detailed features are not confined to specific regions but can progressively propagate into coarser-scale representations; through this step-by-step propagation, the global representation eventually encodes local details. The network includes five key modules—EdgeConv, Encoder, Detail Propagation (DP), Center Sample and Feature FPS. The process is illustrated in Fig. 4. In the following subsections, we describe the component modules (Sec. 4.2.1), and the overall architecture (Sec. 4.2.2).

### 4.2.1 Individual Component Module

*(1) EdgeConv*. We use EdgeConv (Wang et al., 2019) as the local feature extractor in SpreadNet. Its dynamic graph construction captures fine-grained geometric relations among Gaussian primitives, essential for modeling both structural consistency and local variations. Compared to PointNet-style MLPs or global pooling, EdgeConv encodes locality efficiently without the computational overhead of Transformer-based models, providing an effective balance between geometric awareness and efficiency.

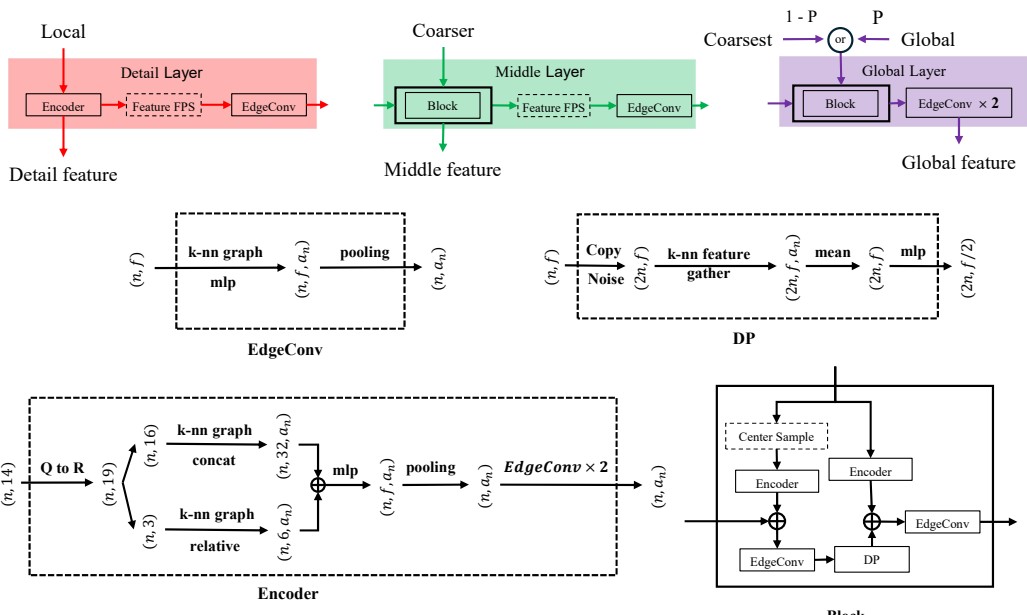

Fig. 4: Overview of the proposed SpreadNet architecture. **Top:** The detail layer, the middle layer, and the global layer. **Bottom:** Key modules, including the EdgeConv, Detail Propagation (DP), Encoder, and the Block structure used in SpreadNet.

*(2) Encoder.* The Encoder extracts point-wise features from Gaussian primitives by decoupling positional and auxiliary attributes. Given the Gaussian set $\mathcal{G}$ as defined in Eq. (1), we precompute rotation matrices $R_i$ from quaternions $q_i$. For the coordinate input $x_i$, spatial relationships are captured via neighborhood differences followed by k-NN max aggregation. For the auxiliary attribute $f_i = [s_i, R_i, \text{sh}_i, \alpha_i]$, we apply the same k-NN max aggregation without differences. The two branches are then fused via MLP and pooling, and further processed through two consecutive EdgeConv layers to produce the final point-wise representation. Formally, the representation is:

$$F_i^{\text{enc}} = \text{EdgeConv}_2(\text{EdgeConv}_1(\text{MLP}(\text{Pool}(concat(h_\theta(x_i, x_j - x_i), h_\phi(f_i, f_j)))))) \quad (5)$$

We compare this encoder design with a simple concatenation baseline in Appendix A.0.3, showing that decoupled representation offers clear performance benefits. Appendix A.0.4 further explores how different attribute aggregations affect performance.

**(3) Detail Propagation (DP).** The Detail Propagation (DP) layer densifies point features while ensuring that local details from lower levels are propagated and coupled with higher-level features, implementing our core hypothesis that local features progressively influence coarser-scale representations. Given input features $F \in \mathbb{R}^{N \times d}$, DP first copies each feature and adds Gaussian noise $\epsilon \sim \mathcal{N}(0, \sigma^2 I)$ to increase the point count, get $F^{copy} \in \mathbb{R}^{2N \times d}$. Neighborhood aggregation is then performed via k-NN mean pooling in feature space, followed by an MLP that projects the aggregated features back to the desired dimension. The entire process can be summarized as

$$F^{DP} = \text{MLP}(\frac{1}{k} \sum_{j \in \mathcal{N}(i)} f_i^{copy}), \quad i = 1, \dots, 2N, \quad (6)$$

where $\mathcal{N}(i)$ denotes the $k$ nearest neighbors of point $i$ in feature space, and $F^{DP} \in \mathbb{R}^{2N \times d/2}$. In this way, DP ensures that higher-level features remain consistent with fine local details, facilitating effective hierarchical feature fusion.

*(4) Center Sample.* To enhance interaction across different scales, we introduce the Center Sample, which selects $n$ Gaussians closest to the origin in normalized space. This operation provides spatially aligned correspondences between lower- and higher-level subsets, facilitating consistent feature propagation while retaining local details. We further report an ablation study on the effect of Center Sample in the appendix A.0.5.

*(5)Feature FPS.* The Feature FPS downsamples Gaussians in the feature space to reduce the number of points while simultaneously expanding their feature dimension. Concretely, we first apply FPS to select a representative subset of features. The selected features are then duplicated and concatenated, effectively doubling the feature dimension. This design cuts computational cost and ensures features' expressiveness and compatibility with adjacent scales, enabling consistent feature propagation across scales. The related ablation experiments were conducted at appendix A.0.6.

### 4.2.2 COMPONENT MODULES OF SPREADNET

The overall architecture of SpreadNet is organized into three hierarchical parts: the detail layer, the middle layer, and the global layer (see Fig. 4). Among them, the middle layer can be repeatedly applied across different scales. To unify the description, we define a Block that encapsulates the shared operations of the middle and global layers. In the detail layer, this Block is replaced by a simpler Encoder, since no lower-level input is available.

Block Structure. Each Block integrates Gaussian primitives $\mathcal{G}^{(k)}$ at the current scale with propagated features $F_{k+1}$ from finer layer. Specifically, an encoder branch extracts structural and semantic features from $\mathcal{G}^{(k)}$, while a propagation branch combines center-sampled Gaussians with $F_{k+1}$, refines them through EdgeConv and DP, and propagates local details upward. The two branches are then fused via EdgeConv to produce the block output $F'_k$, ensuring that scale-specific information is preserved while local details are coupled with global context. See Block in Fig. 4 for details.

**Middle Layers.** For a middle layer, the input consists of the Gaussian primitives at this layer and the propagated features from the finer layer. The Block fuses these inputs to produce the layer representation, which is then simultaneously used for final aggregation across layers and as input to the next layer. This design allows local details and global context to be progressively integrated through the network.

**Global Layer.** Unlike middle layers, the global layer does not propagate features upward and replaces FPS with two consecutive EdgeConv layers to fully encode global information. Its input is stochastically drawn from either the global primitives $\mathcal{G}^{(0)}$ or the coarsest fine-scale primitives $\mathcal{G}^{(1)}$:

$$\tilde{\mathcal{G}}^{(\mathrm{glob})} = \begin{cases} \mathcal{G}^{(0)}, & p \\ \mathcal{G}^{(1)}, & 1-p \end{cases} \tag{7}$$

This random selection prevents the network from over-relying on a single global representation, encouraging balanced integration of global structure and fine-scale details, preventing the global layer from dominating the final decision. Appendix A.0.7 shows that removing this stochastic input (only using $G^{(0)}$) slightly decreases classification performance, confirming the benefit of this design.

**Detail Layer.** For the finest detail layer ($k_{max}$), no finer features are available. Thus, the Block degenerates into a single encoder applied on its primitives. This serves as the starting point of the bottom-up propagation process. For the experiments with different numbers of layers, please refer to Appendix A.0.8.

We note that the design of SpreadNet is inherently generalizable beyond classification. A straightforward extension is to apply a network trained on single-object datasets to segmentation tasks: by fixing the camera and performing a panoramic scan, the network classifies all Gaussians within its view at each step. After completing the scan, a segmentation of the entire scene can be obtained. We provide a detailed demonstration in the Appendix A.0.9.

## 5 EXPERIMENTS

We conduct extensive experiments to evaluate the proposed SPREAD-GS framework. Our study first benchmarks SPREAD-GS against representative baselines on public benchmark datasets, then performs ablation analyses to isolate the effect of each design choice, including sampling strategies, the propagation module, and feature selection. We further assess robustness under different point budgets and field-of-view configurations. This section is organized as follows: Sec. 5.1 reports benchmark results; Sec. 5.2 presents ablations with detailed comparisons. Some comparisons for choices are provided in the Appendix A.0.10.

Table 1: Comparison with point cloud classification baselines on MACGS and ModelNet40GS. SPREAD-GS achieves the best performance on both datasets, showing the effectiveness of multi-scale Gaussian diffusion.

| Method | MACGS | | ModelNet40GS | | Params (M) |
|---|---|---|---|---|---|
| | mAcc (%) | OA (%) | mAcc (%) | OA (%) | |
| PointNet(Qi et al., 2017a) | 84.31 | 85.02 | 85.55 | 90.05 | 3.52 |
| PointNet++ (Qi et al., 2017b) | 89.53 | 89.89 | 87.61 | 90.33 | 1.48 |
| DGCNN(Wang et al., 2019) | 88.93 | 89.52 | 87.40 | 90.98 | 1.81 |
| PointMLP(Ma et al., 2022) | 91.38 | 91.73 | 87.96 | 91.02 | 13.24 |
| DeLA(Yang et al., 2024) | 91.06 | 91.91 | 88.50 | 91.16 | 5.34 |
| Gaussian-MAE (Ma et al., 2025) | 90.67 | 91.73 | 88.87 | 91.55 | 22.10 |
| **SPREAD-GS (ours)** | **93.45** | **93.93** | **89.92** | **91.87** | 2.46 |

## 5.1 BENCHMARK RESULTS

***Data.*** We evaluate SPREAD-GS on two Gaussian Splatting benchmarks. MACGS (Zhang et al., 2025) is a real-world 3DGS dataset with 5,370 samples from 30 categories (4,288 train / 1,082 test). Each object is reconstructed from real captures with Gaussian Splatting, preserving geometry and detail textures. It emphasizes intra-class geometric similarity and inter-class textural variation, making fine-grained and texture-aware recognition challenging. ModelNet40GS (Ma et al., 2025) extends ModelNet40 with 12,489 samples across 40 categories (9,842 train / 2,467 test). Generated from CAD models, it lacks color and textures, focusing on geometric reasoning as a complementary benchmark.

***Baseline***. We prepare a set of representative point cloud classification baselines to fairly evaluate SPREAD-GS. The selected baselines include two categories. Point cloud methods adapted to Gaussian primitives: PointNet (Qi et al., 2017a) and PointNet++ (Qi et al., 2017b) (geometry-driven), PointMLP (Ma et al., 2022) (fully-connected driven), and DGCNN (Wang et al., 2019) and DeLA (Yang et al., 2024) (graph-structure driven). Methods directly designed for 3DGS: Gaussian-MAE (Ma et al., 2025). These data are all obtained by sampling 3DGS at 1024 points, and then concatenating the features to serve as the input.

***Training and testing***. In this experiment, we use $8 \times$ RTX 4090 GPUs. During both training and testing, the 3DGS representation of each object is normalized to a unit sphere. We adopt three detail scales plus a global scale for SPS input, where the FoV values are set to $5°$, $17°$, and $30°$ for the three SPS scales, using FPS (Qi et al., 2017b) for sampling, and the probability $p$ is set to 0.6. For training, we apply data augmentation to all Gaussian primitives, including global scaling (0.8–1.25), random shift ($-0.1$ to 0.1), Gaussian primitive dropout (0.5), and local random jitter (truncated normal distribution with variance 0.01 within $\pm 0.05$). For optimization, we use the AdamW (Loshchilov & Hutter, 2017) optimizer with a cosine annealing (Loshchilov & Hutter, 2016) learning rate schedule, decaying from an initial $10^{-3}$ to $10^{-5}$. The model is trained for 250 epochs with a fixed batch size of 16 and a weight decay of $10^{-4}$. We use a standard cross-entropy loss for training.

***Results.*** On MACGS, which emphasizes fine-grained textures and appearance, SPREAD-GS achieves 93.45% mean class accuracy (mAcc) and 93.93% overall accuracy (OA), surpassing the strongest baseline DeLA by +2.39% mAcc and +2.02% OA (Tab. 1). This demonstrates that our hierarchical propagation effectively propagates local Gaussian details across scales, enabling precise texture-aware classification.

On ModelNet40GS, where geometry dominates and texture cues are weak, SPREAD-GS attains 89.92% mAcc and 91.87% OA, showing a slight improvement over the Gaussian-MAE (+1.05%mACC and +0.32%OA). Importantly, SPREAD-GS significantly reduces the number of parameters (by nearly 10x) compared to Gaussian-MAE, demonstrating its efficiency without compromising performance.

Overall, these results illustrate that SPREAD-GS consistently integrates local details with global structure, performing robustly across tasks with varying emphasis on texture and geometry.

Table 2: Comparison with DGCNN on MACGS. Num denotes input points. ×4 indicates three detail scales and one global scale.

| Method | Num | mAcc (%) | OA (%) |
|--------|-----|----------|--------|
| DGCNN | 1024 | 88.93 | 89.52 |
| DGCNN | 4096 | 90.37 | 90.52 |
| SPREAD-GS | 256×4 | 93.45 | 93.93 |
| SPREAD-GS | 1024×4 | 93.31 | 93.92 |

Table 3: Ablation study on SPS input and propagation on the MACGS dataset. A/P indicates whether features are aggregated (A) or progressively propagated (P) across layers. Each column indicates whether the design choice is used (✓), not used (✗) or not applicable (–).

| Sampling | Network | A/P | DP | mAcc (%) | OA (%) |
|----------|---------|-----|-----|----------|--------|
| FPS | DGCNN | – | – | 90.37 | 90.52 |
| SPS | DGCNN | A | – | 89.64 | 90.33 |
| SPS | SPREAD-GS | P | ✗ | 92.82 | 93.46 |
| SPS | SPREAD-GS | P | ✓ | **93.45** | **93.93** |

## 5.2 ABLATION STUDIES

To assess the contributions of key components in SPREAD-GS, we conduct ablation studies focusing on the Scale-Progressive Sampling (SPS) strategy and the Detail Propagation (DP) module. We adopt DGCNN as a baseline, since SPREAD-GS builds upon its EdgeConv layers; this ensures that observed performance differences primarily reflect the effects of SPS and DU rather than the underlying network architecture. The following analysis highlights the individual and combined impact of these design choices.

Table 2 compares SPREAD-GS with DGCNN under different numbers of input points on the MACGS dataset. Using 1024 points, DGCNN achieves 90.37% OA and 90.52% mAcc. Increasing the point number to 4096 slightly improves DGCNN's performance to 90.52% OA and 90.37% mAcc, with only a minor improvement. In contrast, SPREAD-GS achieves 93.93% OA and 93.45% mAcc with only 256 points per SPS scale (×4 for three detail scales plus one global scale), already surpassing DGCNN with 4096 points. With 1024 points per layer, SPREAD-GS reaches 93.92% OA and 93.31% mAcc. These results indicate that the hierarchical detail scales and the global scale combination in SPREAD-GS effectively capture both coarse structure and fine details, leading to higher classification accuracy with significantly fewer points, and this proves that SPREAD-GS is a more computationally efficient and scalable framework for 3D Gaussian Splatting recognition.

In Table 3, we perform ablation studies on MACGS to quantify the contributions of the Scale-Progressive Sampling (SPS) strategy and the Detail Propagation (DP) module . The baseline uses standard FPS with 4,096 points fed into DGCNN, serving as a point-count–matched reference. Simply concatenating features from the three fine scales and the global layer into a single point set for direct classification slightly reduces performance (-0.73% mAcc, -0.19% OA), indicating that unstructured aggregation disrupts the layer-wise consistency of features. Introducing progressive and layer-wise input into SpreadNet recovers this consistency and significantly improves results (+2.45% mAcc, +2.94% OA), showing the importance of hierarchical processing to capture both global structure and local details. Incorporating the DP module within each Block further boosts accuracy (+0.63% mAcc, +0.47% OA), as it propagates fine-grained Gaussian details across scales, enhancing the integration of structural and appearance cues. Overall, these ablations confirm that SPREAD-GS efficiently leverages multi-scale input while preserving and propagation local details, leading to accurate recognition for both texture-rich and geometry-dominant objects.

## 6 CONCLUSION

In this work, we propose SPREAD-GS, a network designed to more effectively leverage 3D Gaussian Splatting (3DGS) representations, demonstrating clear advantages over directly applying point-cloud-based methods. Our framework integrates a Scale-Progressive Sampling (SPS) strategy to observe objects across multiple scales, capturing both global structure and fine details. We further introduce SpreadNet, which propagates local features to higher levels through hierarchical propagation, enabling multi-scale fusion of structural and appearance information. The experiments on MACGS and ModelNet40GS demonstrate that SPREAD-GS consistently outperforms baselines, validating its ability to exploit both structural and textural information effectively. In addition, the SPS strategy can be applied to other 3D tasks, and SpreadNet may serve as a reference for other model designs.

REPRODUCIBILITY STATEMENT

Our code is released at https://anonymous.4open.science/r/noname-64BE. The framework architecture is fully described in Sec. 4, and training details are provided in Sec. 5.1. All the datasets we used are publicly available, as explained in Sec.5.1.

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

# A  APPENDIX

In this appendix, the parameters of the standard method used for comparison in this method are as follows: 4 layers, each with 1024 points; the field of view angles are 10, 20, and 40; and p is 0.4.

### A.0.1  VALIDATION OF SPS STOPPING CONDITION

To validate the rationality of our sampling stopping criterion, for simplicity, we visualize different sampling levels on the entire lego model (341,297 Gaussians), as shown in Figure 5. From left to right: **A** is the ground truth, representing the most realistic scene; **B** is the unsampled 3DGS rendering, representing the original Gaussian; **C** is the sampling with 100,000 Gaussians, representing the generic case where the sampled set remains on the same order of magnitude and surface details such as protrusions are clearly visible; **D** is the extreme sampling with 34,129 Gaussians, representing a case where surface protrusions are still exactly visible; **E** is the under-sampling with 10,000 Gaussians, representing a case where surface protrusions are no longer visible. From these rendered results, we conclude that maintaining the sampled set within the same order of magnitude (no less than 1/10 of the original) is sufficient to preserve local details.

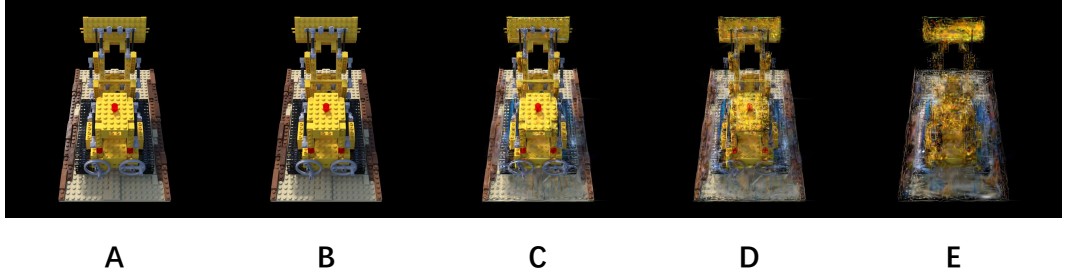

|   A   |   B   |   C   |   D   |   E   |

Fig. 5: This figure visualizes the SPS stopping criterion on a lego and shows how different sampling levels affect the preservation of local geometric details.

### A.0.2  EFFECT OF RANDOM SHIFT IN SPS.

In practice, SPS can introduce a "central bias", where Gaussians near the frustum center are preferentially sampled. To mitigate this, we apply a small random shift to the normalized point cloud before SPS, encouraging the network to capture local details across the entire frustum. To evaluate its effect, we perform an ablation where the random shift is removed. Without this augmentation, SPREAD-GS achieves 91.63% mAcc and 92.46% OA on MACGS, compared to 93.45% mAcc and 93.93% OA with the shift. This demonstrates that the random shift effectively reduces central bias, enabling more uniform coverage of local features and contributing to overall performance.

### A.0.3  ENCODER DESIGN ABLATION

We evaluate the impact of decoupling positional and auxiliary attributes in the encoder. For comparison, we implement a baseline that simply concatenates all features. Results on MACGS are summarized in Table A2. The decoupled design consistently improves both mean class accuracy (mAcc) and overall accuracy (OA), confirming that processing positional and auxiliary attributes separately enhances the encoder's feature quality.

Table A1: Ablation on the random shift augmentation in SPS. ✓indicates the augmentation is applied, ✗ indicates it is not. Removing the shift introduces a central bias, slightly reducing performance.

| Random Shift | mAcc (%) | OA (%) |
|---|---|---|
| ✗ | 91.63 | 92.46 |
| ✓ | 93.45 | 93.93 |

Table A2: Ablation on encoder design. "Concatenation" indicates simple concatenation of positional and other features, while "ours" denotes the proposed decoupled feature encoding.

| Encoder Design | mAcc (%) | OA (%) |
|---|---|---|
| Concatenation | 92.01 | 92.56 |
| ours | 93.45 | 93.93 |

### A.0.4 STUDY ON ATTRIBUTE AGGREGATION

This section provides a detailed analysis of the contribution of different GS attributes to the final accuracy. Specifically, we examine the performance after coupling different attributes and applying the SpreadGS method. The results are shown in the table A3.

In the table, $x$ represents positional information, $\alpha$ denotes opacity, $q$ is the rotation quaternion, $s$ is the scaling matrix, and $sh$ refers to the spherical harmonics used to represent color. In the GS expression, $s$ and $q$ are highly coupled, so there is no need to discuss them separately. From the table, it is evident that using only position and color yields performance nearly as good as using all attributes. However, when $sh$ is combined with other attributes, the performance slightly decreases. When all attributes are included, the performance reaches its optimal value.

Table A3: Contribution of different GS attributes to the object classification accuracy after applying SpreadGS.

| Attributes Used | mAcc (%) | OA (%) |
|---|---|---|
| $x$ | 86.43 | 87.22 |
| $x, \alpha$ | 88.37 | 88.24 |
| $x, sh$ | 92.28 | 93.11 |
| $x, s, q$ | 90.10 | 90.37 |
| $x, \alpha, s, q$ | 90.39 | 91.54 |
| $x, s, q, sh$ | 92.41 | 93.01 |
| $x, \alpha, s, q, sh$ | **93.45** | **93.93** |

### A.0.5 EFFECT OF CENTER SAMPLE

The Center Sample module introduces a slight "central bias" by preferentially selecting points near the origin. However, this spatial alignment ensures that lower- and higher-level features correspond, facilitating consistent feature propagation across zoom layers. Replacing Center Sample with random selection reduces central bias but removes this spatial coupling, slightly lowering performance (Table A4). This highlights that aligned center selection effectively balances detail propagation and layer-wise correspondence.

### A.0.6 EFFECT OF CENTER SAMPLE

The Feature FPS model ensures that when selecting a small subset of features, the original feature information is largely preserved at the feature level. In contrast, replacing it with a random feature selection method results in a loss of completeness in retaining the original feature layer information, which leads to a decrease in accuracy. The experimental results in MACGS are shown in the table A5:

Table A4: Ablation study on the Center Sample module. Random selection replaces the Center Sample.

| Center Selection | mAcc (%) | OA (%) |
|---|---|---|
| Random | 93.20 | 93.84 |
| Center Sample (ours) | 93.45 | 93.93 |

Table A5: Performance comparison between random feature selection and Feature FPS on the MACGS dataset.

| Method | mAcc(%) | OA(%) |
|---|---|---|
| Random Feature | 91.37 | 91.73 |
| Feature FPS | **93.45** | **93.93** |

### A.0.7 EFFECT OF RANDOM INPUT SELECTION IN THE GLOBAL LAYER

We investigate the effect of randomly selecting inputs from the previous layer when forming the global-level input. Specifically, we compare two settings: using the random selection mechanism (Have P) versus directly taking the entire previous layer output without random selection (No P). As shown in Table A6, introducing random selection slightly improves both mAcc and OA, demonstrating its contribution to robustness and generalization.

Table A6: Effect of random input selection in the global layer on MACGS.

| Setting | mAcc (%) | OA (%) |
|---|---|---|
| No P | 93.00 | 93.55 |
| Have P | 93.45 | 93.93 |

### A.0.8 IMPACT OF LAYER DEPTH ON OBJECT CLASSIFICATION ACCURACY

This section examines the impact of the number of layers on object classification accuracy. Specifically, we modify the number of intermediate layers while keeping the total number of sampling points constant. The results are shown below. We observe that the best performance occurs with four layers, which corresponds to two intermediate layers. This can be attributed to the following reasons: when the number of layers is too small, the feature propagation lacks intermediate steps, resulting in a simpler network. This hampers feature diffusion and increases the risk of overfitting. Conversely, if the number of layers is too large, each layer will receive fewer sampling points. This leads to a decrease in the amount of information passed through each layer. As the network becomes deeper under such circumstances, it can result in Table A7,

Table A7: Impact of layer depth on object classification accuracy across different datasets.

| Scales | MACGS | | ModelNet40GS | |
|---|---|---|---|---|
| | mAcc (%) | OA (%) | mAcc (%) | OA (%) |
| 2 | 89.53 | 90.17 | 86.64 | 89.60 |
| 3 | 90.35 | 91.18 | 88.78 | 91.02 |
| **4** | **93.45** | **93.93** | **88.94** | **91.71** |
| 5 | 89.09 | 89.89 | 87.45 | 90.98 |

In addition, we experimented with a setup that combines multiple detail layers with a global layer, aiming to reduce reliance on randomly selected details from a single object. However, this configuration led to a similar issue as with fewer layers, where the lack of sufficient intermediate steps hindered detail propagation, resulting in a decrease in accuracy. This result is extremely similar to the accuracy rate without the DP. The specific results are shown in the table A8:

Table A8: Performance comparison between the three Detail layers and a Global layers setup and SpreadGS method.

| Method | mAcc(%) | OA(%) |
|---|---|---|
| Detail*3 + Global | 92.29 | 93.47 |
| SpreadGS | **93.45** | **93.93** |

### A.0.9 A SEGMENTATION OF THE ENTIRE SCENE

In this section, we validate the conjecture regarding the segmentation method presented in the paper. The proposed approach is naturally well-aligned with the perspective, making it suitable for perspective-based scanning. By classifying objects within a given viewpoint and performing multiple perspective samplings of the scene, we can achieve an approximate segmentation result. Specifically, we randomly select four objects from the MACGS dataset and arrange them separately to form a multi-object scene. Subsequently, perspective sampling is applied, where objects within each sampled perspective are classified under the assumption that all objects within the same viewpoint belong to the same category. By performing multiple perspective samplings within a single scene, we obtain the results shown figure 6. The left image shows the composed scene, while the right image displays the segmentation mask generated after processing, with different colors representing distinct object categories. It is important to note that the generated mask is not a conventional 2D mask. Instead, we produce a 3D mask based on 3D GS points, which provides an advantage for downstream tasks that require more detailed segmentation masks.

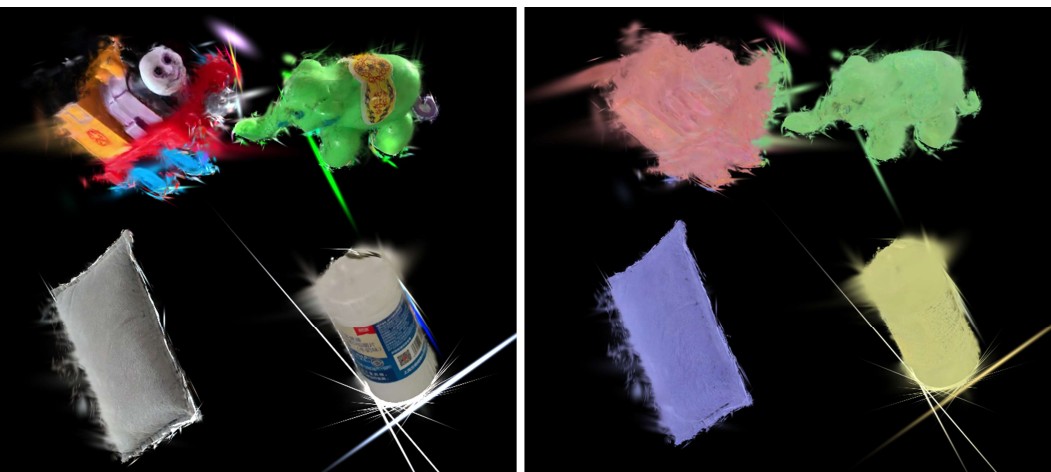

Fig. 6: This figure demonstrates the feasibility of the segmentation. The left shows the created multi-object scene, and the right shows the generated 3D mask.

### A.0.10 EFFECT OF DIFFERENT FIELD-OF-VIEW (FOV) CONFIGURATIONS

We study the impact of different FoV settings in the Scale-Progressive Sampling (SPS) on MACGS. Table A9 shows the details. Specifically, we change the number of FoV angles used in the progressive sampling: 5°, 10°, and 20°;5°, 17°, and 30°; 10°, 20°. During this process, we observed an interesting phenomenon: when the number of sampling points per layer is reduced, selecting an appropriate FoV angle ensures that the results remain nearly unchanged. By reducing the number of sampling points per layer, it is possible to maintain performance by simply narrowing the FoV angles, ensuring that the GS (geometrically sampled) points within each viewpoint retain sufficient detailed information. This indirectly demonstrates the strong stability of our network.

Table A9: Effect of different FoV configurations on MACGS.

| FoV Angles | num | mAcc (%) | OA (%) |
|---|---|---|---|
| 5, 10, 20 | 1024*4 | 92.23 | 92.90 |
| 5, 17, 30 | 256*4 | 93.45 | 93.93 |
| 10, 20, 40 | 1024*4 | 93.31 | 93.92 |

### A.0.11 USE OF LARGE LANGUAGE MODELS

We used ChatGPT (OpenAI) to assist with language polishing, editing,and LaTeX formatting. The model was not involved in the conception of methods, implementation, or experimental design. All scientific decisions and the intellectual content of this work were made exclusively by the authors.

