# OpenReview forum: "SPREAD-GS: Scale-Progressive Representation Extraction and Detailing for 3D Gaussian Splatting"
_ICLR.cc/2026/Conference — ICLR 2026 Conference Withdrawn Submission_

### Official Review · Reviewer_uFqF · 2025-10-17

**Soundness:** 3
**Presentation:** 3
**Contribution:** 3
**Rating:** 4
**Confidence:** 3

**Summary:**

In this paper, the authors propose a framework for object classification, named SPREAD-GS. Its key module SPS generate multi-scale subsets and by progressively narrowing the camera FoV, and then SpreadNet encodes these subsets and propagating details across scales through noise-augmented feature upsampling. The experiments show that SPREAD-GS achieves the best results on the MACGS and ModelNet40GS datasets.

**Strengths:**

1. The authors propose leveraging Gaussian attributes as additional information to enhance the object classification capability, which I consider effective and innovative.
2. The experiments demonstrate that SPREAD-GS achieves superior performance on the MACGS and ModelNet40 datasets.

**Weaknesses:**

1. Incorporating more Gaussian attributes increases the input storage size compared to other methods that use point cloud representations. It would be valuable if the authors could provide additional experiments to validate the model’s performance under conditions where the input storage size is comparable across different methods. This would help clarify whether the performance improvement of SPREAD-GS stems from the richer Gaussian attributes or merely the larger input size.
2. As the authors claim that Gaussian attributes enhance the object classification capability. To better support this claim, it is recommended that the authors present comparative experiments to demonstrate the impact of combined Gaussian attributes—such as scale, rotation, opacity, and color—versus using only position information.
3. The design of SPS focuses on single object. However, multi-object interactive scenes are more common in practical Gaussian Splatting applications(e.g., autonomous driving). It would be better if the authors could provide more examples or experiments on multi-object scenes to evaluate SPREAD-GS’s performance in handling object interactions.

**Questions:**

The definition of the camera pose during SPS remains unclear, which causes confusion. As an object’s appearance varies with the observation direction, the distance and angle between the camera and the object can significantly affect the sampling results. Additionally, the robustness of SPS to different camera poses also raises confusion.

---

> ### Author Response · Authors · 2025-11-24
>
> We appreciate the reviewer's insightful suggestions and agree with the strengths you identified in our paper. After carefully reading the points you raised in the Weaknesses and Questions sections, we will address each point as follows.
>
> Weaknesses1: About Input
>
> We appreciate the reviewer’s concern regarding the increase in input storage size due to the incorporation of more Gaussian attributes. To clarify, our baseline methods also use all Gaussian attributes (e.g., position, scale, rotation, opacity, and color) as input. Therefore, we can confidently assert that the improved performance is not merely due to increased input dimensionality and memory usage but is rather a result of the differences in network architecture and sampling strategy. Additionally, we have updated the baseline section (page 8) in the manuscript to better explain this point.
>
> Weaknesses2: Impact of Gaussian Attributes
>
> Thank you for highlighting this important aspect. We have now conducted a comprehensive ablation study to evaluate the impact of different Gaussian attributes on classification performance. The results are summarized in the table below:
>
> | Attributes Used     | OA (%) | mAcc (%) |
> |---------------------|--------|----------|
> | C                   | 87.22  | 86.43    |
> | C, o                | 88.24  | 88.37    |
> | C, sh               | 93.11  | 92.28    |
> | C, s, r             | 90.37  | 90.10    |
> | C, o, s, r          | 91.54  | 90.39    |
> | C, s, r, sh         | 93.01  | 92.41    |
> | C, o, s, r, sh      | 93.93  | 93.45    |
>
> In this study, we did not separate scale (s) and rotation (r) as they are deeply coupled in the Gaussian representation, and separating them was not necessary for our analysis. These experimental results and further discussion have been added to Section 4.2.1(2) Encoder (page 6) and Appendix A.0.4 (page 13).
>
> Weaknesses3: Multi-Object Scene Handling
>
> Thank you for pointing this out. Indeed, in the original manuscript, we briefly mentioned the theoretical application of our method to multi-object segmentation but did not provide experimental evidence. To address this, we created a scene with 4 objects and applied the panoramic scanning method described in the paper to perform a simple segmentation task. The results are shown in Figure 6 (page 15), where we observe that the generated masks are 3D-based, rather than the conventional 2D masks, providing greater precision for tasks requiring fine details. This experiment demonstrates the potential for applying our method to multi-object segmentation. We have updated this in Appendix A.0.9 (page 15) with the new visual results and discussion.
>
> Questions: Camera Pose in SPS
>
> We appreciate the reviewer’s insightful question regarding the camera pose during the Scale-Progressive Sampling (SPS) process. As noted in the manuscript, the camera pose plays an essential role in selecting the appropriate field of view for sampling specific details of the scene. It acts as a tool for focusing on different levels of detail and linking global and local information across scales. During training, we randomly change the camera view to expose the model to various perspectives and details. At test time, the model is also fed with random camera views, which is reflected in our code. This randomness helps avoid overfitting to any particular viewpoint and ensures robust generalization.
>
> Please kindly let us know if you have any further concerns.

---

### Official Review · Reviewer_fxNS · 2025-10-31

**Soundness:** 3
**Presentation:** 3
**Contribution:** 3
**Rating:** 6
**Confidence:** 2

**Summary:**

This paper introduces SPREAD-GS, a new framework that aims to enhance 3D object classification using 3D Gaussian Splatting (3DGS) representations. The authors argue that existing approaches either rely on 2D features embedded into 3DGS, which fail to exploit native 3D information, or treat Gaussians as point cloud, which discards fine-grained per-primitive details. To address this, SPREAD-GS proposes two key components: (1) Scale-Progressive Sampling (SPS): A multi-scale sampling mechanism that progressively narrows the FoV to create hierarchical subsets of Gaussians. It captures both global geometric structure and localized appearance details. (2) SpreadNet: A hierarchical network that processes these subsets and propagates fine-scale features across scales via a Detail Propagation (DP) module. The network leverages EdgeConv layers for local structure encoding and fuses features progressively to build robust object-level representations. Experiments on MACGS and ModelNet40GS benchmarks show that SPREAD-GS outperforms state-of-the-art point cloud and Gaussian-based baselines.

**Strengths:**

1. Good writing and organization.

2. Effective idea with clear motivation.

3. The proposed scale-progressive sampling and SpreadNet are novel and with solid engineering.

**Weaknesses:**

1. Lack of efficiency report. While SPREAD-GS is presented as efficient, the paper does not report detailed runtime, memory usage, or FLOPs. Given the multiple hierarchical scales and EdgeConv-based architecture, quantitative efficiency comparisons with baselines (e.g., DGCNN, PointMLP) would strengthen the claim of practical implementation and scalability.

2. According to Figure 3, although SPREAD-GS adopts random offsets to alleviate the central bias, during the Scale-Progressive Sampling process, each time it extracts Gaussians from a smaller region, it still focuses only on a localized area instead of sampling a large number of local features across multiple smaller regions. This may lead to a significant bias in local feature extraction.

3. The component ablation of SpreadNet is not sufficiently fine-grained. Although the paper performs ablations on hierarchical feature aggregation and propagation, it does not conduct step-by-step ablations on its five major modules, including EdgeConv, Encoder, Detail Propagation, Center Sample, and Feature FPS, to verify the necessity and contribution of each component.

**Questions:**

See the weaknesses.

---

> ### Author Response · Authors · 2025-11-24
>
> Thank you for your suggestions. Your summary is very accurate and it truly highlights my strengths. Moving forward, I will respond to each of the weaknesses you mentioned one by one.
>
> Weaknesses1: Efficiency Report
>
> Thank you for pointing this out. Indeed, efficiency is a key aspect of our work, and one of the main advantages of SPREAD-GS over Gaussian-MAE. Our model reduces the number of parameters by nearly 10x compared to Gaussian-MAE. This reduction in parameters is one of the key strengths of our method. Below is a comparison of model sizes, and we have updated the baseline and results in experiments (page 8) and some additional expressions were added to the abstract (page 1), introduction (page 2), and experimental section (page 8).
> | Model | Params (M) |
> |-------------|------------|
> | PointNet | 3.52 |
> | PointNet++ | 1.48 |
> | DGCNN | 1.81 |
>  | PointMLP | 13.24 |
> | DeLA | 5.34 |
>  | Gaussian-MAE| 22.10 |
>  | SPREAD-GS | 2.46 |
>
> Weakness 2: Local Feature Sampling Strategy
>
> Thank you for your insightful suggestion. The approach you proposed was indeed an alternative that we initially considered. However, after careful evaluation, we decided not to pursue this strategy. In this method, we implemented three detail layers, and then propagated their details to the global layer, in an attempt to simulate sampling a large number of local features.
>
> We verified this alternative method, and it resulted in an accuracy of 93.47% (OA) and 92.29% (mAcc), which is slightly lower than our original approach (−0.46% OA and −0.16% mAcc). We believe this alternative lacks the hierarchical progression required for maintaining proper detail propagation. Our results show that it performs similarly to the case where the Detail Propagation (DP) module is absent, which achieved 93.46% OA and 92.28% mAcc, thus confirming our belief. We have updated Appendix A.0.8 (page 14 and page 15)to include these things.
>
> Weakness 3: Component Ablation Study
>
> Thank you for pointing this out. Due to space limitations, we did not include all component ablation results in the main text. We focused on the Detail Propagation (DP) module in the main manuscript because it is the most critical component of our method. However, we have included the ablation results for other modules in the appendix. The table below summarizes the results:
>
> | **MACGS**              | **mAcc**  | **OA** | **Location in Manuscript**                |
> |------------------------|---------|----------|-------------------------------------------|
> | Full model             | 93.45   | 93.93    | -                                         |
> | Without DP             | 92.82   | 93.46   | Section 5.2, page 9                      |
> | Encoder concat         | 92.01   | 92.56    | Appendix A.0.3, pages 12--13             |
> | Center sample random   | 93.20   | 93.84    | Appendix A.0.5, pages 13--14             |
> | Feature FPS random     | 91.37  | 91.73    | Appendix A.0.6, pages 13--14             |
>
> Please kindly let us know if you have any further concerns.

---

> > ### Comment · Reviewer_fxNS · 2025-11-28
> >
> > Thanks for your response. I think my core concerns are addressed.

---

> > > ### Author Response · Authors · 2025-11-28
> > >
> > > Thanks for your timely response, and we really appreciate your insightful suggestions to our work!

---

### Official Review · Reviewer_hkUC · 2025-10-31

**Soundness:** 2
**Presentation:** 2
**Contribution:** 2
**Rating:** 4
**Confidence:** 4

**Summary:**

This paper introduces SPREAD-GS, a method for classifying 3D Gaussians. It leverages SPS to sample 3DGS at multiple scales, feeding each scale into the proposed SpreadNet to extract corresponding features. The final global feature representation is then used for classification.

**Strengths:**

1.Demonstrates strong classification performance on datasets such as MACGS and ModelNet40GS, validating the method’s effectiveness.

2.Proposes SpreadNet with novel encoder and block designs, showing clear architectural innovation.

3.Integrates multi-scale 3DGS features, enhancing attention to local details and improving their contribution to classification accuracy.

**Weaknesses:**

1.The method is primarily focused on 3D Gaussian classification. Although Section 4 suggests potential application to semantic segmentation, no experiments are provided, leaving its effectiveness on segmentation tasks uncertain. This may limit its broader applicability.

2.The ablation study lacks evaluation of the impact of varying the number of detail scales and the use of Feature FPS on classification performance.

3.While the paper references "Mitigating Ambiguities in 3D Classification with Gaussian Splatting (CVPR 2025)" and uses its datasets, no direct comparisons with that method are reported.

**Questions:**

The visual results are quite limited, only one lego case is provided. Should provide more visual results of used benchmarks.

---

> ### Author Response · Authors · 2025-11-24
>
> We are extremely grateful for your suggestions regarding this work. I have carefully read your review and thank you for your positive assessment of the merits of our research. Subsequently, I will respond to the Weaknesses and Questions:
>
> Weakness 1: Lack of Experimental Validation for Segmentation
>
> As you correctly pointed out, merely mentioning the theoretical potential without experimentation is not sufficient to make a convincing case. Therefore, in the meantime, we conducted feasibility experiments based on the approach mentioned in Section 4. We have made revisions to the main text at the end of Section 4 (page 7) and added Appendix A.0.9 (page 15) to provide additional experimental validation. In this part, we created a scene containing 4 objects and applied the panoramic scanning approach described in Section 4. The model classified the Gaussians in each view, and we colored each object differently to illustrate the segmentation mask. As shown in Fig. 6 (page 15), our method produces a “3D mask,” as opposed to 2D-rendered masks, allowing for more accurate segmentation in 3D space. This experiment demonstrates that, even when trained for classification, the model naturally adapts to segmentation tasks through its panoramic scanning mechanism.
>
> Weakness 2: Ablation Study on number of  Scales and Feature FPS
>
> We have conducted an ablation study on the number of detail scales. Instead of simply adding or removing layers, we fixed the total number of Gaussians and varied the number of Gaussians per scale. The results are shown in the table below:
>
> | Scales | MACGS OA(%) | MACGS mAcc(%) | ModelNet40 OA(%) | ModelNet40 mAcc(%) |
> |----------|----------|------------|---------------|-----------------|
> | 2        | 90.17    | 89.53      | 89.60         | 86.64           |
> | 3        | 91.18    | 90.35      | 91.02         | 88.78           |
> | 4  | **93.93**  | **93.45**  | **91.71**     | **88.94**       |
> | 5        | 89.89    | 89.09      | 90.98         | 87.45           |
>
> We found that the best performance was achieved with 4 scales, and there was no significant improvement when the number of scales was increased further. We have updated this in the end of Section 4.2.2 Detail Layer (page 7), as well as in Appendix A.0.8 (page 14).
>
> Interestingly, during this process, by modifying the minimum field of view at the fourth layer, we achieved better results than before. This indicates that adjusting the minimum field of view range helps to reduce the number of samples at the finest scale while still maintaining the accuracy of classification. This analysis is included in Appendix A.0.10 (page 15 and page 16). We have updated the best version of the text, including the abstract (page 1), introduction (page 2), and experimental section (page 8).
>
> Additionally, we further investigated the effect of Feature FPS by comparing it with random feature selection. The results are as follows:
>
> | Method         | MACGS OA(%)  | MACGS mAcc(%) |
> |----------------|----------|------------|
> | Random Feature | 91.73    | 91.37      |
> | Feature FPS    | **93.93** | **93.45**  |
>
> These results confirm that Feature FPS is essential for improving performance. We have included this discussion at the end of Section 4.2.1 (Feature FPS) on page 7 and in Appendix A.0.6 (pages 13 and 14).
>
> Weakness 3: Reference to Mitigating Ambiguities in 3D Classification with Gaussian Splatting (MACGS)
>
> We appreciate the reviewer’s reference to MACGS, which is primarily a **dataset-driven** work, introducing the MACGS dataset and demonstrating that GS can be treated as an augmented version of point clouds for classification. However, MACGS did not propose a corresponding method. This is the reason why we only used the dataset from this study.
>
> Question: Limited Visual Results
>
> In response to your comments, we have added more visual examples of our model’s performance on different objects, enhancing the clarity and diversity of our visual results. Specifically, we replaced the "lego" example in Figure 1 (page 2) with a "chair" example to show the versatility of the model across different objects.
>
> Please kindly let us know if you have any further concerns.

---

### Official Review · Reviewer_JS8v · 2025-10-31

**Soundness:** 2
**Presentation:** 2
**Contribution:** 1
**Rating:** 4
**Confidence:** 4

**Summary:**

The paper proposes SPREAD-GS, a novel framework for 3D object classification using native 3D Gaussian Splatting (3DGS) representations, introducing Scale-Progressive Sampling (SPS) to generate multi-scale Gaussian subsets and SpreadNet, a hierarchical network with Detail Propagation (DP) to fuse local details and global structure across scales.
Evaluated on MACGS (texture-rich) and ModelNet40GS (geometry-only), SPREAD-GS achieves 93.92% (+2.01% over SOTA) and 91.67%, respectively, with comprehensive ablations validating the efficacy of SPS, DP, and encoder design.

**Strengths:**

The paper presents a well-motivated and timely approach to 3D object classification using native 3D Gaussian Splatting representations, a largely underexplored direction.
Its originality lies in the thoughtful combination of multi-scale sampling (SPS) and cross-scale detail propagation (DP), which creatively adapts hierarchical reasoning to the unique structure of 3DGS beyond treating it as a plain point cloud. The technical quality is high, with comprehensive experiments on two distinct benchmarks (MACGS and ModelNet40GS) and thorough ablations validating each component. The clarity of presentation is strong, with clear figures and logical flow, and the significance is notable—by demonstrating that 3DGS contains rich semantic cues in its native attributes, the work opens new pathways for 3D understanding tasks beyond reconstruction.

**Weaknesses:**

1 Questionable task motivation: 3D Gaussian Splatting (3DGS) is primarily a reconstruction output rather than a standard input modality, and it is rarely the starting point for classification in real-world pipelines, making the practical relevance of this task unclear.

2 Unfair comparison: The comparison with Gaussian-MAE is conducted without using its intended pretraining setup—the paper explicitly trains it from scratch, which undermines the fairness and validity of the comparison.

**Questions:**

Fairness of comparison with Gaussian-MAE: Gaussian-MAE is designed as a large-scale self-supervised pretraining framework for 3DGS, where downstream tasks like classification and segmentation are used to evaluate the quality of its pretrained features—not as standalone task-specific models. Training Gaussian-MAE from scratch (without pretraining) for classification, as done in this work, does not reflect its intended use and undermines the comparison.
Since the proposed SPREAD-GS is trained from scratch and tailored specifically for classification, comparing it against a general-purpose pretrained backbone under a non-standard (scratch) setting is neither fair nor meaningful.

---

> ### Author Response · Authors · 2025-11-24
>
> We are extremely grateful for your recognition of the advantages of our work. Your summary of the strengths of the paper is very accurate. Next, we will respond to each of the weaknesses and questions:
>
> Weakness 1: Motivation
>
> While 3D Gaussian Splatting (3DGS) was originally proposed as a rendering representation, recent work has demonstrated its growing adoption as a geometric and appearance representation for 3D scene understanding tasks, beyond mere rendering. For instance, methods such as ShapeSplat, Gaussian Grouping, and MACGS explore segmentation, detection, and representation learning directly on 3DGS. These works suggest that 3DGS is emerging as a promising input modality for 3D perception tasks.
>
> However, existing methods typically (1) treat Gaussians as an extension of point clouds, using point cloud networks without fully leveraging the unique properties of 3DGS, or (2) rely on large-scale pre-trained models for scene inference. These approaches do not design networks specifically tailored to understanding the distinctive attributes of 3DGS objects. In contrast, our work proposes a network architecture explicitly designed to understand and exploit the unique properties of 3DGS, contributing to the emerging field of 3DGS-based perception.
>
> Weakness 2 and Questions: Fairness of Comparison with Gaussian-MAE
>
> We appreciate the reviewer’s insightful comment regarding the comparison with Gaussian-MAE. You are correct that Gaussian-MAE is intended as a pretraining framework, and evaluating it from scratch does not reflect its intended usage. Our original comparison aimed to maintain equal data and supervision budgets across all methods, but we agree that neglecting the pretraining stage of Gaussian-MAE renders the comparison incomplete.
>
> To address this, we conducted additional experiments where Gaussian-MAE was first self-pretrained on the corresponding dataset (MACGS or ModelNet40GS) using its masked-reconstruction objective, followed by fine-tuning for classification. The updated results are shown below:
>
> | Method                            | Pretraining? | MACGS mAcc(%) | MACGS OA(%)  | ModelNet40 mAcc(%)  | ModelNet40 OA(%)  |
> |-----------------------------------|--------------|------------|----------|-----------------|---------------|
> | Gaussian-MAE             | No           | 90.21      | 90.63    | 87.86           | 90.61         |
> | Gaussian-MAE  | Yes          | 90.67      | 91.73    | 88.87           | 91.55         |
> | SPREAD-GS          | No           | **93.45**  | **93.93**| **88.94**       | **91.71**     |
>
> From these results, we observe that:
> 1. Pretraining does indeed improve Gaussian-MAE's performance, and we have now included this pretraining setting in the paper to ensure a fairer comparison.
> 2. Even with pretraining, Gaussian-MAE’s performance still falls short compared to SPREAD-GS on both datasets.
> 3. SPREAD-GS achieves strong results without any pretraining, highlighting the effectiveness of our architecture in leveraging native 3DGS attributes.
>
> We updated the content of the second row in the table to the relevant Gaussianmae content in the baseline section (page 8), and modified all the parts that compared with SOTA in the abstract (page 1), introduction (page 2), and experimental section (page 8).
>
> Please kindly let us know if you have any further concerns.

---

> > ### Comment · Reviewer_JS8v · 2025-11-24
> >
> > Thank you for your response.
> >
> > What I would like to clarify is that evaluating a pretrained Gaussian-MAE model fine-tuned on your downstream task would be much more meaningful. Since large pretrained models typically serve as stronger backbones for 3D perception tasks, it is important to demonstrate whether your method can effectively leverage such pretrained knowledge.
> >
> > If your approach cannot benefit from a large pretrained backbone and cannot outperform methods that do, then the strength of your contribution becomes limited.
> > Therefore, pretraining only on the test dataset does not provide meaningful evidence of generalization or practical value.
> >
> > Given the above issues, I maintain my rating of 4.

---

> ### Author Response · Authors · 2025-11-27
>
> Thank you for your patience and thoughtful feedback.
>
> We have added results for the Gaussian-MAE model fine-tuned on the corresponding datasets (MACGS and ModelNet40) after using the official pretrained weights. The results are summarized in the table below, where "no pretrain" represents the performance of Gaussian-MAE without any pretraining, "self" represents the results after pretraining on the respective dataset and fine-tuning on the same dataset, and "official" refers to fine-tuning using the official pretrained weights on the respective dataset.
>
> | Method                | MACGS mAcc(%) | MACGS OA(%) | ModelNet40 mAcc(%) | ModelNet40 OA(%) |
> |-----------------------|---------------|-------------|--------------------|------------------|
> | Gaussian-MAE (no pretrain) | 90.21         | 90.63       | 87.86              | 90.61            |
> | Gaussian-MAE (self)        | 90.67         | 91.73       | 88.87              | 91.55            |
> | Gaussian-MAE (official)      | 90.86         | 91.91       | 89.11              | 91.83            |
> | SPREAD-GS (old)            | 93.45         | 93.93       | 88.94              | 91.71            |
> | SPREAD-GS                  | **93.45**         | **93.93**       | **89.92**              | **91.87**            |
>
> During training, we adjusted some of the hyperparameters of SPREAD-GS on ModelNet40, which led to improved results on this dataset. As shown in the table, the results obtained with the official pretrained weights show a slight improvement over those obtained with self-pretraining, but still do not surpass our method. Additionally, as we do not rely on pretraining and large models, SPREAD-GS still offers an advantage in terms of memory efficiency compared to Gaussian-MAE.
>
> We have updated the relevant sections related to the SOTA comparison in the manuscript, specifically in the Introduction (page 2) and Experimental section (page 8).

---

### Note · Authors · 2026-03-05

I have read and agree with the venue's withdrawal policy on behalf of myself and my co-authors.

---

### Meta-Review · Area_Chair_sgPx · 2026-01-07

**Summary:**

The submission received mixed ratings from four reviewers. Three of them provided negative ratings. They have several major concerns that are closely related to the key contribution of the paper, including unclear motivation illustration, unfair experimental comparison, and missing model efficiency analysis. The rebuttal from the authors only partially addressed these issues. Based on the overall ratings and comments from the reviewers, AC finally decided to recommend a rejection of this submission.

**Reviewer Concerns:**

There are several major concerns from the reviewers: (i) unclear motivation illustration, (ii) unfair experimental comparison, and (iii) missing model efficiency analysis. The rebuttal provided by the authors only partially addressed these issues, as echoed by the reviewers.

**Reviewer Scores:**

The original score distribution is 6, 4, 4, 4. After the rebuttal, the reviewers did not indicate any intention to raise their scores.

---

### Decision · Program_Chairs · 2026-01-26

Reject